# Unraveling the regulative development and molecular mechanisms of identical sea urchin twins

Haruka Suzuki[1], Junko Yaguchi [1], Koki Tsuyuzaki [2,3,4,5] &
Shunsuke Yaguchi [1,2] ✉

Since Hans Driesch's pioneering work in 1891, it has been known that animal embryos can develop into complete individuals even when divided. However, the developmental processes and molecular mechanisms enabling this self-organization remain poorly understood. In this study, we revisit Driesch's experiments by examining the development of isolated 2-cell stage blastomeres in the sea urchin, *Hemicentrotus pulcherrimus*. Contrary to intact embryos, these isolated blastomeres initially form a flat, single layer of dividing cells that eventually round up to be a blastula. Live imaging and knockdown experiments reveal that actomyosin activity at the basal side of the cells and septate junctions drives this process. Intriguingly, we observed temporal disorganization of the anterior-posterior (A-P) and dorsal-ventral (D-V) axes, where the original A-P poles come into contact after sphere shape formation. The disrupted A-P axis is subsequently corrected as the embryos employ the Wnt/β-catenin signaling mechanisms assumed to be used in intact embryos to re-establish a normal axis. These findings suggest that axis re-organization through pre-existing developmental mechanisms is essential for the successful regulative development of divided embryos.

In 1891, Hans Driesch demonstrated that isolated 2-cell and 4-cell stage sea urchin blastomeres can each develop into a complete individual, highlighting the remarkable self-organizing capacity of early embryos[1]. Following this seminal discovery, researchers observed similar self-organization abilities in early embryos across various taxa[2–6]. Even in humans, this phenomenon underpins the formation of identical twins when early embryos split accidentally. In contrast, animals relying on mosaic development, where maternal factors predetermine cell fate during early stages, lack this self-organization capacity (e.g., comb jellyfish, solitary ascidians, tusk shells)[7].

Despite Driesch's experiments being referenced in textbooks for over a century, the molecular mechanisms governing this self-organizing ability remain poorly understood. Moreover, detailed morphological and developmental analyses of separated early embryos are scarce (Fig. 1a). Echinoderms, especially sea urchins, offer an ideal model system to study self-organization due to their external fertilization, ease of manipulation, and transparency—making them suitable for high-resolution live imaging. Additionally, the extensive genomic and gene regulatory data available for sea urchins further supports their utility in unraveling these mechanisms[8–12]. Remarkably, sea urchins were the original model system used by Driesch to demonstrate self-organization. While there is one report on isolated blastomere development in sea stars[13], detailed studies on sea urchins remain elusive, and this gap must be addressed.

Here, we revisit Driesch's classic experiments using modern molecular biology techniques to define the self-organization phenomenon in

[1]Shimoda Marine Research Center, University of Tsukuba, Shizuoka, Japan. [2]PRESTO, Japan Science and Technology Agency, Tokyo, Japan. [3]Department of Artificial Intelligence Medicine, Graduate School of Medicine, Chiba University, Chiba, Japan. [4]Institute for Advanced Academic Research (IAAR), Chiba University, Chiba, Japan. [5]Laboratory for Bioinformatics Research, RIKEN Center for Biosystems Dynamics Research, Wako, Saitama, Japan. ✉e-mail: yag@shimoda.tsukuba.ac.jp

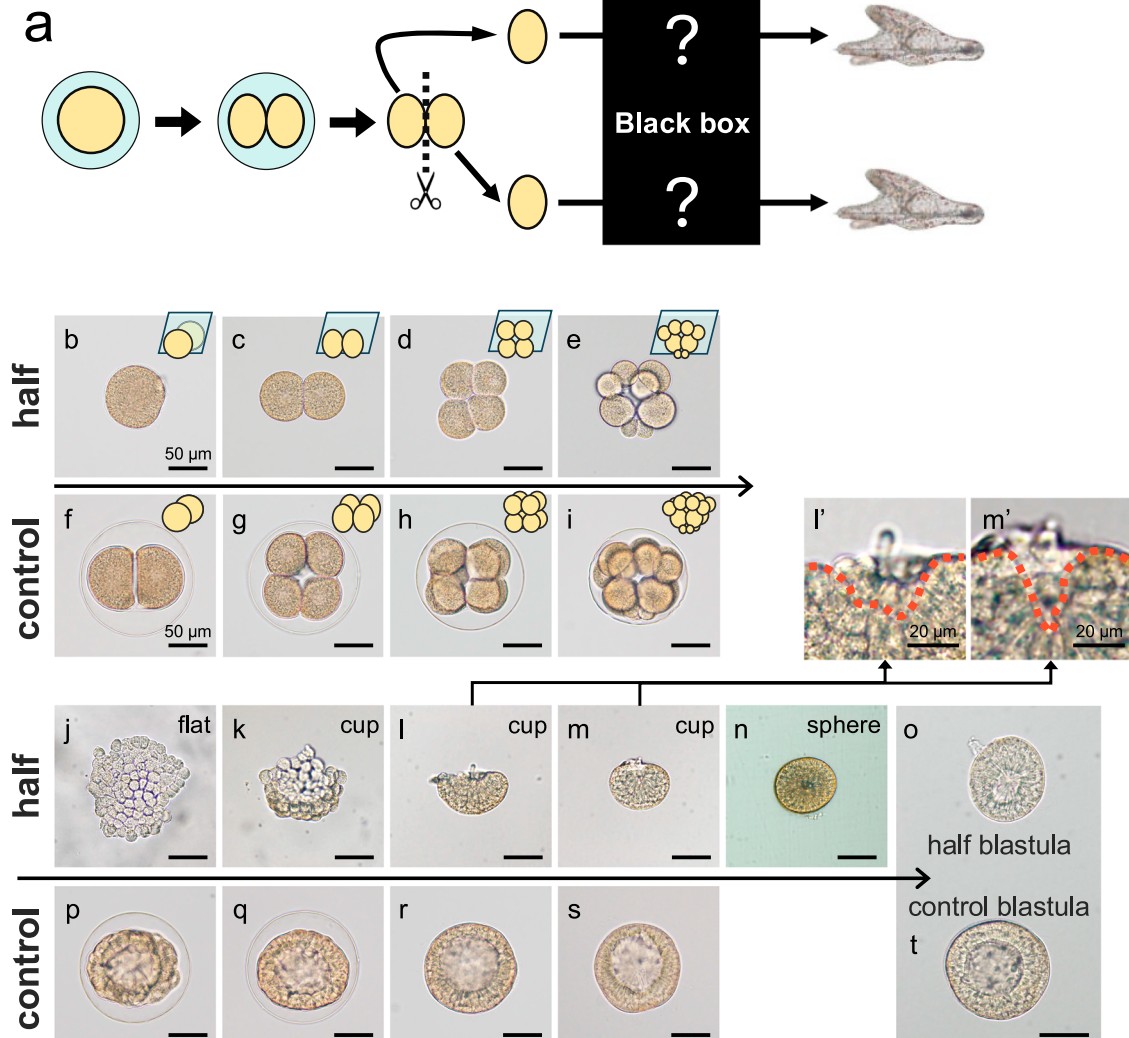

**Fig. 1 | Halved *H. pulcherrimus* embryos undergo unique developmental process. a** Schematic image of experiment and question of this research. Eggs were fertilized, then 2-cell stage embryos were divided into half after removing fertilization envelope. **b–t** Developmental processes of halved embryos (**b–e**, **j–o**) and the corresponding intact embryos (**f–i**, **p–t**). Developmental stages specific to halved embryos were named as flat, cup and sphere. (**l′**, **m′**) Magnified images of cup opening site in (**l**, **m**). Images shown are representative of nine independent experiments performed with similar results. Orange dashed lines, outline of edge of the cup shape embryos. Scare bars, 50 μm (full images), 20 μm (**l′**, **m′**).

sea urchins. Focusing on the Japanese species *Hemicentrotus pulcherrimus*, we describe the developmental trajectory of isolated 2-cell stage blastomeres, with particular attention to the molecular forces driving morphological changes. Additionally, we report the re-organization of the anterior-posterior (A-P) and dorsal-ventral (D-V) axes during the development.

## Results

### Halved *H. pulcherrimus* embryos undergoes unique developmental process

Initially, we examined the morphology of *H. pulcherrimus* embryos bisected at the 2-cell stage (Fig. 1). During early cleavage, each isolated blastomere followed a cleavage pattern similar to that of intact embryos (Fig. 1b–i). For example, at the 16-cell stage, when intact embryos formed 8 mesomeres, 4 macromeres and 4 micromeres, halved embryos generated 4 mesomeres, 2 macromeres and 2 micromeres. However, after the 16-cell stage, while intact embryos continued cleaving to form a blastula (Fig. 1p–s), the halved embryos unexpectedly developed a flat, plate-like structure instead (Fig. 1j). The edges of this flattened structure then lifted, forming a cup-like shape (Fig. 1k), which gradually closed at the opening (Fig. 1l, m), eventually

creating a sphere (Fig. 1n). At this stage, the blastocoel was difficult to discern in the spherical embryos. Ultimately, the halved embryos developed a clear blastocoel (Fig. 1o), becoming miniature versions of an intact blastula (Fig. 1t).

Given the essential role of the extracellular matrix (ECM), hyaline layer, which we removed during blastomere isolation, in normal sea urchin development[14,15], we considered whether its absence caused the irregular morphology. However, this possibility was ruled out, as isolated blastomeres with an intact hyaline layer exhibited the same morphological changes (Supplementary Fig. 1). This indicates that the observed developmental process is an intrinsic feature of bisected embryos. We named these distinct stages of halved embryo morphology as flat, cup and sphere (Fig. 1j–n). Although the development of halved embryos progressed more slowly than intact embryos, once the blastula was formed, the subsequent developmental processes—such as mesenchyme cell ingression, gastrulation, and pluteus formation—proceeded in a manner similar to that of intact embryos (Supplementary Fig. 2).

During our observations, we identified two key biological questions regarding the developmental re-organization of halved embryos. First, how does the flat morphology transition into a

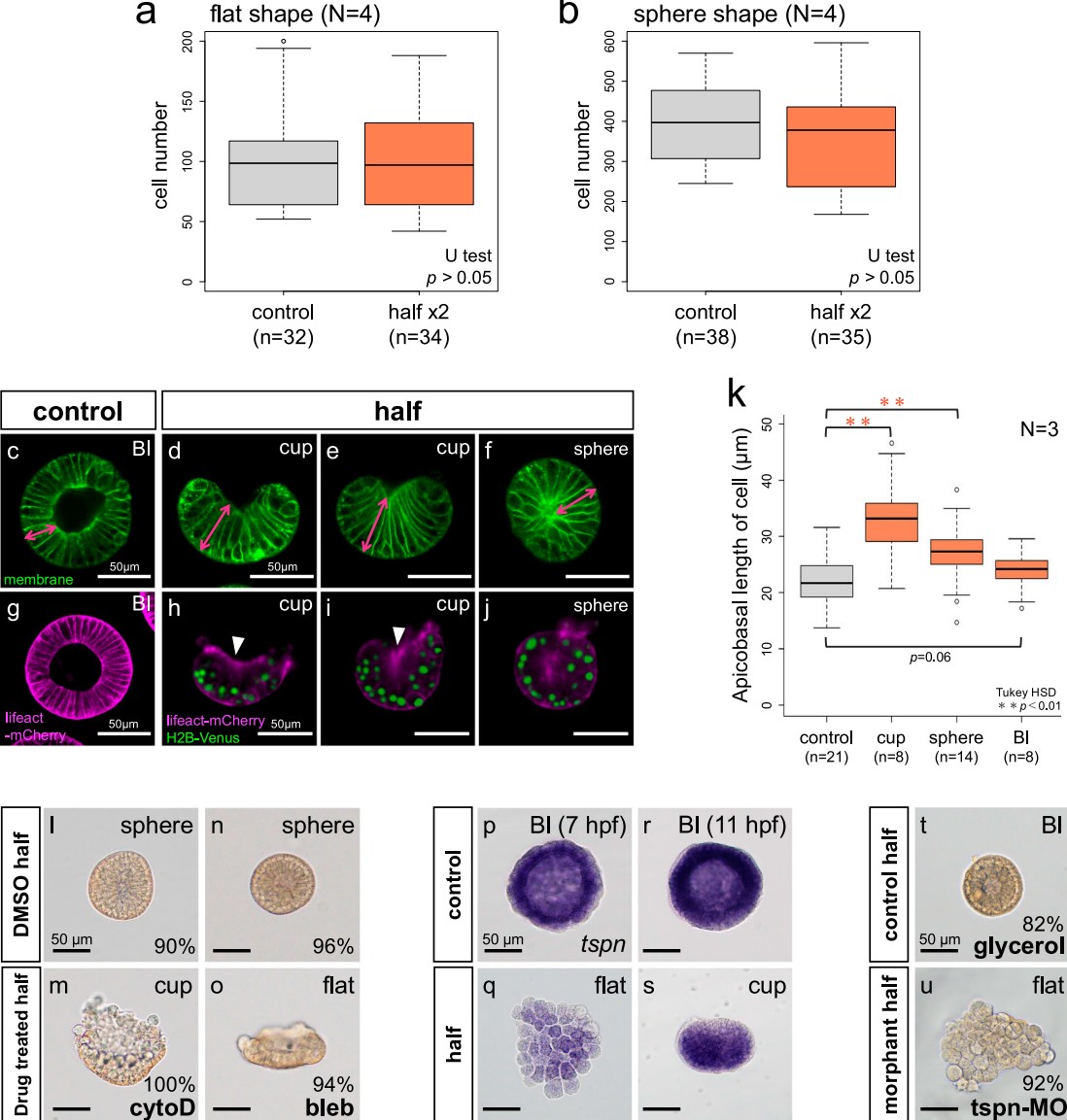

**Fig. 2 | Actomyosin and septate junction are key factor for forming accurate sphere shape. a, b** Cell number comparison between intact embryos and double-counted halved embryos at flat and sphere shape stage. Statistical significance was determined using the two-sided Wilcoxon rank-sum test. [**a** control vs half x2: *p* value = 0.88, *N* = 4, **b** control vs half x2: *p* value = 0.087, *N* = 4]. **c–f** Cell membrane was stained with FM1-43. Magenta double-sided arrows indicate cell length along apicobasal axis (*N* = 3). **g–j** Actin and nuclei live imaging with lifeact-mCherry mRNA and histone2B-Venus mRNA in halved embryos. White arrowheads indicate actin accumulation sites. Results are based on three succeeded live-imaging. **k** Cell length comparison along apicobasal direction between each stage of halved embryos and control (*N* = 3). Statistical significance was assessed using a one-way ANOVA followed by Tukey's Honestly Significant Difference (HSD) test for multiple comparisons. All tests were two-sided and adjusted for multiple comparisons

(control vs cup: *p* = 0.001, control vs sphere: *p* = 0.001, control vs Bl: *p* = 0.061). **l–o** Actin polymerization and myosin II activity inhibition in halved embryos with cytochalasin D (cytoD) and (-)-blebbistatin (bleb), respectively [*N* = 2 in (**l**, **m**) and *N* = 3 in (**n**, **o**)]. **p–s** Gene expression pattern of septate junction related gene, tetraspanin (tspn) (*N* = 1). **t, u** Morphology of control halves and tspn morphant halves (*N* = 3). Percentage in images indicates the ratio of individuals that showed the same morphology with pictures at the timing of observation. Developmental stage is written in the upper right in each image. Bl blastula, hpf hours post fertilization. All box plots show the median (center line), the 25th and 75th percentiles (bounds of the box), and the minimum and maximum values excluding outliers (whiskers). Outliers are plotted as individual points. Scale bars, 50 μm. Source data are provided as a Source Data file.

sphere? Second, how is a normal body axis established in the resulting sphere, given that the original axis could be disrupted during this shape transformation?

**Actomyosin and septate junction contribute the shape transition of halved embryos**
To investigate the mechanism driving the flat-to-sphere transition, we examined whether this morphological change was accompanied by additional cell proliferation. Our analysis showed no significant

difference in cell number between intact embryos and double-counted halved embryos, indicating that the shape change is not due to increased cell proliferation but rather a result of cell shape changes (Fig. 2a, b). To further understand these cell shape dynamics, we closely observed the transition from cup to sphere using the membrane probe FM1-43 (Fig. 2c–f). Similar to intact blastulae, both the cup-shaped and sphere-shaped halved embryos were composed of a single layer of cells. However, the cells in halved embryos were notably more elongated along the apical-basal axis compared to those in intact

embryos, forming a distinct cone-like shape (indicated by magenta double-headed arrows in Fig. 2c–f, k). The cell length in cup-shaped halved embryos was significantly greater than that in control embryos at 8 hours post fertilization (hpf) (corresponding to the flat stage of halved embryos), 10 hpf (cup stage equivalent), and 14 hpf (blastula stage equivalent) (Supplementary Fig. 3). Therefore, cell elongation along the apicobasal axis appears to be a specific feature of halved embryos. As development progressed, the apices of these cone-shaped cells converged at the center of the spherical embryo, while the edges of the cup gradually sealed laterally, leading to complete closure. During the subsequent transition from sphere to blastula, the elongated cells shortened to adopt a columnar shape, similar to those found in intact blastulae (Fig. 2k). This series of coordinated cell shape changes underlies the morphological re-organization from a flat structure to a spherical blastula-like form in halved embryos.

To identify the factor(s) regulating the drastically morphological transition from flat to spherical shape, we first perturbed several signaling pathways known to play key roles in embryonic development. However, none of the treatments affected the shape transition (Supplementary Fig. 4). This suggested that the morphological change in halved embryos might be driven by cell-autonomous mechanisms. We therefore focused on the cytoskeleton and performed live imaging of actin dynamics in halved embryos by injecting lifeact-mCherry mRNA (Fig. 2g–j, Supplementary Movie 1). To visualize the nuclei and determine the apicobasal axis, we co-injected histone2B-Venus mRNA, as nuclei localize to the apical side of cells in normal sea urchin blastulae[16]. In control blastula, actin accumulation was observed both apical and basal side and the accumulation at apical side is stronger than that at the basal side (Fig. 2g). Unexpectedly, in halved embryos, we observed strong actin polymerization on the basal side of the cells, opposite the nuclei, during the cup-to-sphere transition (Fig. 2h, i, white arrowheads). To determine if this basal actin polymerization is essential for the morphological change, we treated halved embryos with cytochalasin D, an inhibitor of actin polymerization, and (-)-blebbistatin, an inhibitor of myosin-II ATPase, from the flat stage (Fig. 2l–o). While over 90% of control halves successfully formed a spherical shape, 100% of cytochalasin D-treated halves were arrested at the cup stage and never formed spheres. Additionally, 94% of blebbistatin-treated embryos exhibited significant delays in sphere formation, resulting in incomplete spherical blastulae (Supplementary Fig. 5). In both treatments, cell elongation along the apicobasal axis was not observed. These results suggest that actomyosin-generated forces on the basal side of cells are crucial for cell elongation along the apicobasal axis, and this cell shape change is essential for the proper formation of the blastula.

To identify additional factors involved in the shape transition of halved embryos, we examined the role of septate junctions, which are septa-like structures between cells that function as occluding junctions in invertebrates[17]. In *Strongylocentrotus purpuratus*, septate junctions are formed at the 8th cleavage stage[18]. Notably, the timing of the flat-to-cup shape transition in halved embryos coincided with the initiation of septate junction formation, prompting us to investigate the role of septate junction-related genes, particularly Tetraspanin[19]. in situ hybridization revealed that *tetraspanin* was ubiquitously expressed across the entire embryo in the flat stage, and this expression persisted through the sphere stage (Fig. 2p–s, Supplementary Fig. 6a, b). This expression pattern was identical to that in control embryos. To assess whether septate junctions are required for the shape transition in halved embryos, we attenuated tetraspanin function using a morpholino anti-sense oligo (MO) and analyzed the resulting phenotypes (Fig. 2t, u, Supplementary Fig. 6c, d). While 82% of control halved embryos successfully transitioned to the sphere stage, only 8% of tetraspanin-deficient halved embryos achieved this transition. Most tetraspanin-deficient halved embryos exhibited a "bumpy" flat shape, where cells were misaligned and

failed to form a single cell layer. By 24 hpf, a small percentage of tetraspanin-deficient embryos eventually adopted a spherical shape. A similar phenotype was observed in halved embryos where ZO-1, another component of septate junctions[19], was knocked down using a specific morpholino (Supplementary Fig. 6d). These results strongly suggest that the mechanical forces provided by septate junctions contribute significantly to the shape transition from flat to spherical in halved embryos.

Based on these observations, we identified two molecular mechanisms regulating the shape transition in halved embryos: actomyosin constriction and the adhesive force of septate junctions. Surprisingly, neither mechanism relies on the signaling pathways typically associated with body axes formation. Instead, shape transition depends on an intrinsic system. The cells themselves actively adhere to each other via septate junctions, and this strong cellular adhesion facilitates the morphological change from a flat to a spherical structure as the cells reorganize to form a cohesive, unified shape.

### Re-organization of the anterior-posterior axis in halved embryos through re-utilization of endogenous posterior signals

Next, to characterize the gene expression feature of halved embryos, we performed bulk RNA-seq and compared the gene expression profile between intact embryos and halved embryos by principal component analysis (PCA) (Fig. 3a). Halved embryos were sampled at 4 stages: cup, sphere, 6-8 hours after sphere formation and early gastrula. Intact embryos were sampled at 2-cell stage and corresponding timing with each stage of halved embryos. At cup shape stage, the gene expression profile of control and halved embryos were similar. At the sphere stage, a large difference was detected between control and the halves. Surprisingly, the difference between control and halves was rapidly compensated 6-8 hours after the sphere shape formed. This result implies that some critical event for accomplishing self-organization occurred after sphere formation. We especially focused on genes which is involved in body axes formation and compared gene expression amount of 5 anterior specification genes (foxQ2, homeobrain, six3, sFRP1/5, dkk-1), 4 posterior specification genes (wnt8, wnt1, frizzled5/8, blimp1) and 4 lateral ectoderm specification gene (nodal, lefty, chordin, bmp2/4) between control and halves, and the only detectable difference was foxQ2 expression amount (at cup stage: cup/control = 4.63). Therefore, we focused on foxQ2 and traced the anterior-posterior (A-P) axis formation in halved embryos. We analyzed gene expression patterns in halved embryos using in situ hybridization chain reactions with *foxQ2* (an anterior marker) and *alx1* or *foxA* (posterior markers) (Fig. 3b–f). In intact embryos, *foxQ2* and *foxA* were expressed at opposite poles, and halved embryos initially exhibited similar patterns with control (Fig. 3b–d). However, immediately after the halved embryos formed a spherical shape, *foxQ2* and *foxA* were expressed in close proximity, almost adjacent to each other (Fig. 3e), indicating a complete disruption of the A-P axis. Remarkably, six hours later, the expression sites of *foxQ2* and *foxA* had returned to opposite poles, resembling normal A-P axis (Fig. 3f). These observations show that the A-P axis temporarily collapses when the spherical shape was formed but is subsequently reorganized to restore normal axis patterning.

Since no aberrant cell proliferation was observed during the formation of the spherical shape (Fig. 2a, b), we hypothesized that A-P axis re-organization occurs through one of three possible mechanisms: (1) shifting the anterior end specification site from its original to its final position, (2) shifting the posterior end specification site from its original to its final position, or (3) shifting both the anterior and posterior end specification sites to their final positions. To determine the most likely scenario, we labeled the animal-pole cells of halved flat-stage embryos with DiI (Fig. 3g) and tracked the labeled cells at the gastrula stage to determine their location in the body. In intact sea urchin embryos, animal-pole cells typically contribute to the anterior end

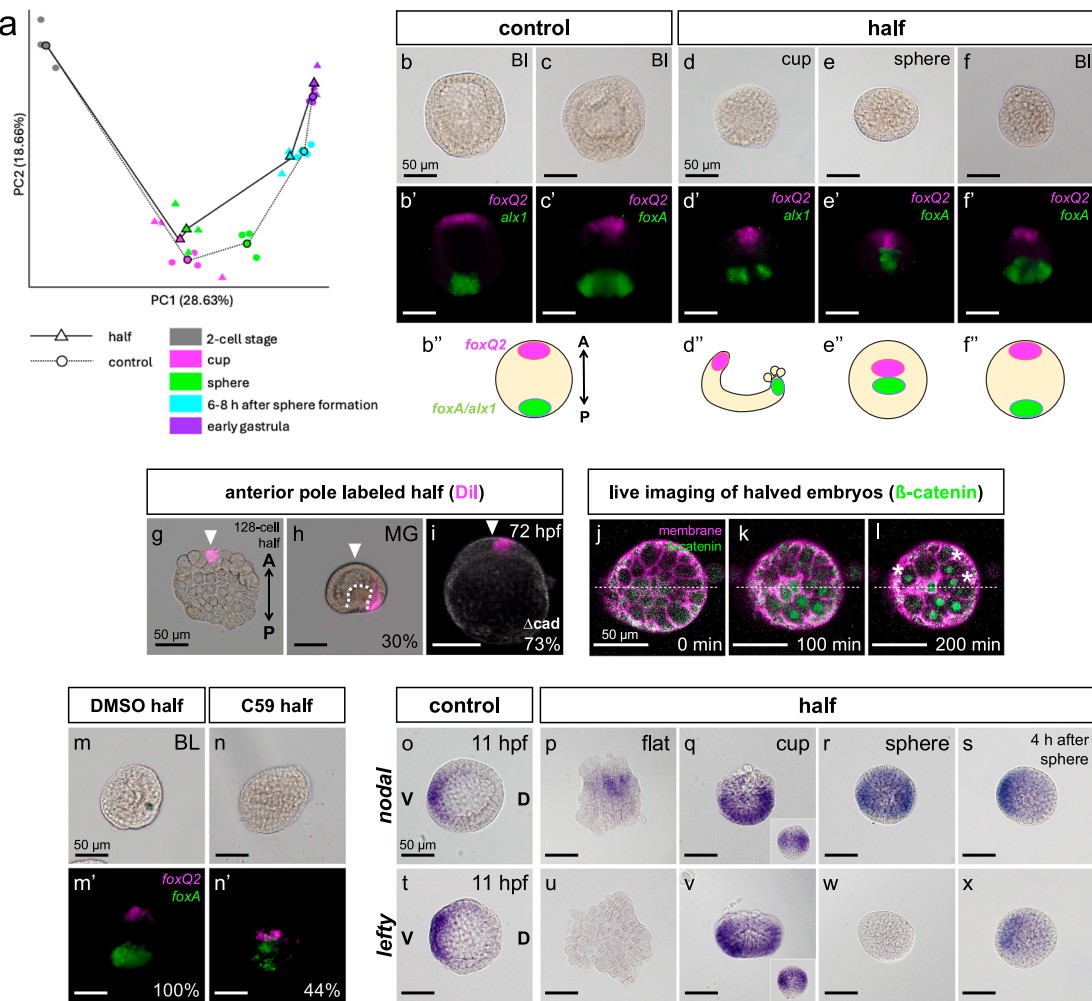

**Fig. 3 | Re-organization of anterior-posterior and dorsal-ventral axes in halved embryos. a** PCA analysis in halved embryos (triangles) and control (circles). Centroids of each stage were connected by line. **b–f** in situ hybridization chain reaction of intact embryos and halved embryos with *foxQ2* as anterior marker and *alx1* or *foxA* as posterior marker. b" and d"-f" show the schematic image of positional relationship between anterior and posterior region. d", lateral view; e", cup rim view; f", lateral view. A-P, anterior-posterior axis. Representative images from at least three independent experiments with similar results are shown. **g–i** Animal pole cell was labeled with DiI (**g**), then the labeled cell position was observed at gastrula stage (**h**). Same experiment was performed with Δcadherin mRNA injected halves (**i**). Percentages in (**h**, **i**) indicate the ratio of individuals whose labeled cell observed at the same position with picture [*N* = 8 in (**h**) and *N* = 6 in (**i**)]. Detail results of analysis is written in Supplementary Fig. 7. White arrowheads, position of anterior pole; White dashed line, outline of the archenteron. **j–l** Live-imaging of ß-

catenin-Venus mRNA and membrane-TagRFP mRNA injected halves. Representative images from at least three independent experiments with similar results are shown. White dashed lines, border of animal and vegetal half; asterisks, ß-catenin positive nuclei in animal half. **m**, **n** in situ hybridization chain reaction with *foxQ2* and *foxA* in control halved and Wnt/ßcatenin inhibited halves with C59. Percentage in (**m'**, **n'**) indicate the ratio of individuals that showed the same positional relationship between *foxQ2* and *foxA* expression site with pictures (*N* = 2). **o–x** Gene expression patterns of *nodal* and *lefty* in intact and halved embryos. Both lateral view (q, v) and cup rim view (inserted images in q, v) were observed at cup shape. Images shown are representative of ten independent experiments performed with similar results. V-D, ventral-dorsal axis. Developmental stages are written in the upper right in each image. Bl blastula, MG mid gastrula, hpf hours post fertilization. Scale bars, 50 μm. Source data are provided as a Source Data file.

region[12]. However, in 78.8% of halved embryos, the labeled cells were found in the lateral ectoderm, suggesting that the anterior end was specified in a different position from the original. Remarkably, in 30% of the halved embryos, the labeled cells were located in the posterior lateral ectoderm (Fig. 3h, Supplementary Fig. 7). We also performed the same labeling experiment on posterior cells of halved embryos (Veg2 lineage, which give rise to secondary mesenchyme cells and archenteron cells[12]). In 75% of halved embryos, the labeled cells contributed to the archenteron or became secondary mesenchyme cells at the gastrula stage (Supplementary Fig. 7), indicating that the posterior region was specified at its original site. These results suggest that A-P axis re-organization is primarily accomplished through the shift of the anterior end specification site from its original to its final position, while the posterior end maintains its original position.

To investigate whether the Wnt/β-catenin, which is the first and the most responsible signaling for A-P patterning[20,21], plays a role in the shifting of the anterior end specification site in halved embryos, we blocked the pathway by injecting mRNA encoding a truncated form of cadherin's cytoplasmic domain (Δcad). This truncated cadherin prevents β-catenin from entering the nucleus and functioning as a transcription factor[22,23]. We labeled the animal pole cells of the halved Δcad embryos with DiI to trace the cell fate shift. As a result, the labeled cells remained at the most anterior end, indicating that the animal-pole cells retained their original fate (Fig. 3i [cf. Fig. 3g, h]). This result strongly suggests that endogenous Wnt/β-catenin signaling is necessary for the anterior end shift in halved embryos. Intriguingly, at the timing of the anterior-fate shifting in halved embryos, Wnt/β-catenin originally does not play a role in restricting the anterior end region in intact embryos.

Thus, we hypothesized that Wnt/β-catenin signaling is reactivates irregularly in halved embryos at this stage. To test this, we performed live imaging of halved embryos injected with β-catenin-Venus mRNA, which allowed us to trace where Wnt/β-catenin signaling was active[22,24]. In these embryos, we observed a temporal nuclear β-catenin signals in a region adjacent to the original posterior site after the halved embryos formed a sphere (white asterisks in Fig. 3j–l, Supplementary Movie 2). To confirm whether canonical Wnt/β-catenin signaling is truly reactivated after the embryo has reached the sphere stage—at a time point when it would normally be too late to suppress *foxQ2*, we treated embryos with the Wnt inhibitor C59 from cup shape stage and analyzed the expression patterns of *foxQ2* and *foxA* at blastula stage. While C59 does not achieve the same level of canonical Wnt inhibition as Δcadherin injection in *H. pulcherrimus*, it has a relatively strong inhibitory effect and was thus used in this experiment. In control halved embryos, 100% (16/16) showed polar expression of *foxQ2* and *foxA* at opposite ends (Fig. 3m, m'), consistent with successful A–P reformation. In contrast, 44% (7/16) of C59-treated halved embryos exhibited *foxQ2* and *foxA* expression in adjacent regions, indicating a failure of *foxQ2* to shift to the opposite pole after initial proximity to *foxA* after being formed sphere (Fig. 3n, n'). Overall, these results indicate that a transient Wnt/β-catenin signal plays a crucial role in shifting the anterior end region from its original to the final position, facilitating the re-organization of the A-P axis in halved embryos following sphere formation. In addition, non-canonical Wnt signaling has been reported to restrict anterior neuroectodermal fate downstream of canonical Wnt in normal sea urchin embryos[20,25], we next investigated whether this pathway is involved in anterior fate shifting in halved embryos. We inhibited c-Jun N-terminal kinase (JNK), a key component of non-canonical Wnt signaling, using the JNK inhibitor SP600125, and observed the positional relationship between *foxQ2*-positive anterior regions and *foxA*-positive posterior regions in halved embryos (Supplementary Fig. 8). In 97% of control halves, the anterior and posterior regions were located at opposite poles, similar to intact embryos. However, in 27% of JNK-inhibited embryos, the anterior and posterior regions were adjacent to each other, suggesting that non-canonical *W*nt signaling is partially required for anterior end shifting. These findings suggest that signaling pathways, which normally restrict the anterior-most region in intact embryos, are also utilized during A-P axis re-formation in halved embryos.

## Dorsal-ventral axis is reorganized during halved embryos development

Considering the organization of body axes, we next focused on the dorsal-ventral (D-V) axis. During sea urchin embryogenesis, D-V axis formation is regulated by members of the TGF-β superfamily, including Nodal, Lefty, Chordin, and BMP2/4[26–28]. To determine whether the D-V axis forms normally in halved embryos, we performed in situ hybridization with *nodal* and *lefty* (Fig. 3o-x). In intact embryos, *nodal* is expressed exclusively in the future ventral ectoderm. In halved embryos, although the precise expression position was challenging to completely identify in the flat stage, we detected biased *nodal* expression. Intriguingly, from the cup stage onward, *nodal* was ubiquitously expressed across the entire embryo, except at the anterior and posterior ends, this pattern that persisted until the spherical stage. Four hours after the halved embryos formed a sphere, *nodal* expression became re-biased toward the future ventral side, resembling that of intact embryos. We also observed nuclear phosphorylated-Smad2/3 (pSmad2/3), a downstream transcription factor of the Nodal signaling pathway[29], throughout the embryo at the cup stage, except at the anterior and posterior ends (Supplementary Fig. 9), supporting the notion that ectopically expressed Nodal functions as a signaling ligand. The expression pattern of *lefty* closely mirrored that of *nodal* during the cup stage, though it was less prominent immediately after the embryos reached the sphere stage; however, the precise mechanism responsible for this transient reduction remains undetermined in the current study (Fig. 3w). To investigate BMP signaling, we performed immunohistochemistry to detect phosphorylated-Smad1/5/8 (pSmad1/5/8), a transcription factor downstream of BMP[29], which marks the region where BMP2/4 functions (Supplementary Fig. 10). In halved embryos, pSmad1/5/8 signals were first observed at 16 hpf, coinciding with the timing in intact embryos, and were localized on one side, as in normal embryos. Notably, no pSmad1/5/8 signals were detected before the embryos reached the sphere stage (Supplementary Fig. 10g). These findings suggest that the D-V axis is initially present in halved embryos and is later reorganized after the spherical shape is established.

## Discussion

Our data revealed that *H. pulcherrimus* 2-cell halved embryos undergo a unique developmental process, progressing through flat, cup and sphere stages (Figs. 1, 4a). This specific sequence had never been reported in any sea urchin species, making it an intriguing discovery for us. We also observed this developmental pattern in other sea urchins, such as *S. purpuratus* and *Strongylocentrotus intermedius*, when embryos were halved at the 2-cell stage (Supplementary Fig. 11). In contrast, 2-cell blastomeres of *Temnopleurus reevesii* developed directly into a blastula without passing through the flat or cup stages. A key difference in *T. reevesii* development is the lack of compaction until the 60-cell stage, while other echinoderms exhibit strong cell-cell adhesion from early cleavage stages[30]. Further studies with additional echinoderm species will help elucidate whether this developmental process is completely common across the phylum. Interestingly, previous research demonstrated that 2-cell stage sea urchin embryos, in which one blastomere was killed while maintaining the membrane structure, could still form a blastula[31] (Supplementary Fig. 12a-f). This led to the hypothesis that sea urchin embryos "actively" recognize their individual boundaries through cell-cell signal communication[31]. At first glance, the pronounced morphological changes from the flat or cup to the sphere in our study might suggest an 'active' reorganization of individual cell boundaries, contributing to the formation of the spherical blastula body through cell-cell signaling. However, no major signaling pathways were involved in this shape transition and there is no difference between control and halves in gene expression profile at cup shape stage (Fig. 3a and Supplementary Fig. 4). Therefore, we propose that the establishment of embryonic boundaries during development is governed primarily by cell-autonomous mechanisms in halved embryos, rather than by active cell–cell signaling. The observed morphological outcome—the formation of a spherical blastula—appears to emerge as a cumulative result of individual cells autonomously regulating their own shape and behavior. In this view, the overall halved embryo shape is not sculpted by coordinated, long-range intercellular signaling, but rather by the local, self-organized activities of individual cells acting independently, yet coherently.

In sea urchin and sea star eggs, the first cleavage establishes apical-basal polarity, where the apical side adheres to the ECM and the basal side to other cells. This polarity is maintained even when 2-cell stage blastomeres are separated[32–35], indicating that apical-basal polarity maintenance is a cell-autonomous event. Therefore, the characteristic flat shape of halved embryos likely results from the apical nature of the cell, as the apical side continues to adhere to the ECM even in isolated blastomeres. Our data suggest that septate junctions play a crucial role in the transition from flat to sphere shape in halved sea urchin embryos (Fig. 2p–u). Septate junctions are rapidly formed following cell reaggregation, as seen in hydra regeneration from dissociated cells[36], and dissociated cells from sea urchin blastulae can also regenerate a whole organism from cell aggregates[37,38]. Although the precise mechanisms of septate junction assembly in sea urchin embryos remain unclear, it is plausible that this process is cell-autonomously initiated in early development. In addition, halved embryos never fuse to other halves after they make sphere. Thus, we

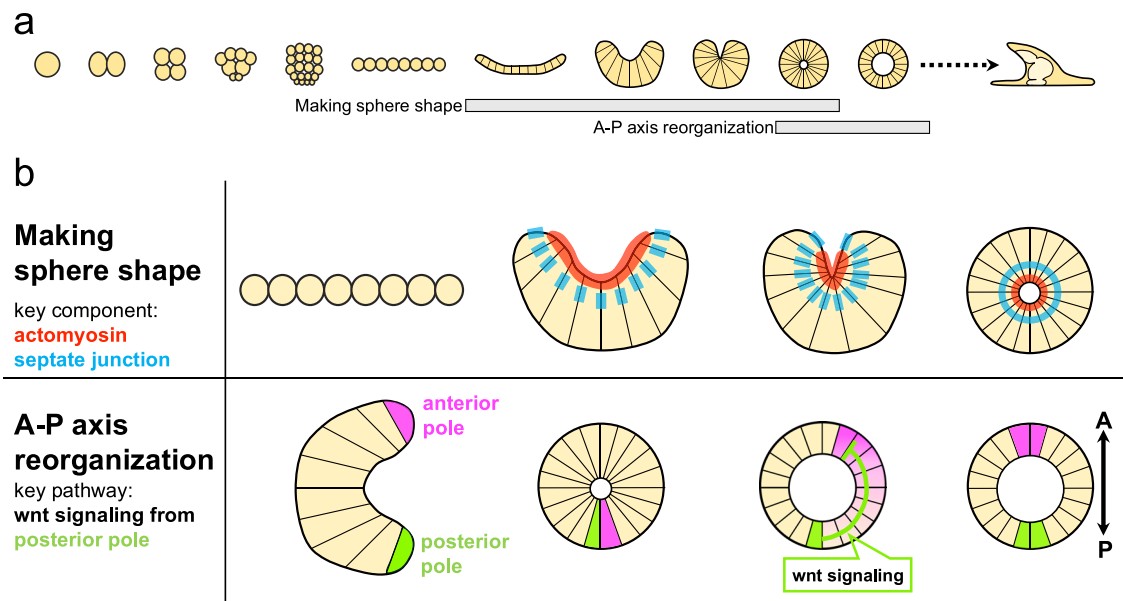

**Fig. 4 | Self-organization process in sea urchin twins. a** Schematic summary of halved *H. pulcherrimus* development. Grey bars indicate the time frame that each event which we argued in this article occurs. **b** Two crucial events in sea urchin self-organization which we discussed. (Upper row) Making sphere shape: At the timing for shape transition from flat to cup, actomyosin activity at the basal side of the cells (red) contribute to cells elongate along apicobasal axis direction and make cells cone chape. Apex of each cell is bundled and becomes the center of the embryo. Simultaneously, septate junction (blue) gives each cell the ability to adhere each other on the lateral side. As a result, the free edge of the cup can bind each other and form sphere. (Lower row) A-P axis re-organization: At the cup shape stage, presumptive anterior pole (magenta) and posterior pole (green) are in their original positions. When sphere shape is formed, presumptive anterior and posterior region are temporarily placed very close together and anterior-posterior (A-P) axis is temporarily disturbed. Triggered by anterior and posterior pole contact, Wnt/ß-catenin signal is newly activated at adjacent region of original posterior pole, and the signal makes the anterior-most region shift to the opposite side. Finally proper A-P axis is formed.

propose that septate junctions are part of a cell-autonomous mechanism determining the individual boundary in halved embryos. The lateral regions of cells are predisposed to form septate junctions with neighboring cells, enabling the flat/cup structure to close the opening and form a single-layered sphere as septate junctions develop between the cells at the edge of the halved embryos (Fig. 4b). We also suggest that actomyosin activity at the basal side of cells contributes to the dynamic morphological changes in halved embryos by facilitating cell shape alterations, although we cannot completely rule out a potential contribution from apical actin. Since extreme cell elongation along the apical-basal axis was not observed in halved sea star embryos[6,13], nor in other sea urchin species (Supplementary Fig. 11), we hypothesize that the cell elongation observed in *H. pulcherrimus* halved embryos may be a species-specific feature. In addition, it is implied that actin accumulation on basal side of the cell is halved embryos specific response because actin accumulation at the apical side was stronger than the basal side in normal blastula (Fig. 2g). We cannot definitively determine whether the accumulation of actin at the basal side is cell-autonomous. However, it is implied that halved embryos utilize some intrinsic mechanisms to regulate the actin accumulation and cell shape change because halved embryos could make sphere with any signaling perturbation (Supplementary Fig. 4).

This research revealed that the temporarily disrupted A-P axis during the flat-to-sphere transition is gradually restored afterward. The re-organization of the A-P axis is triggered by the proximity of the anterior and posterior regions, with the anterior end shifting from its original position, driven by vegetal signals temporarily activated by Wnt/β-catenin (Fig. 4b). This study provides the first experimental evidence of the molecular mechanism underlying axis re-formation during self-organization. As shown in Fig. 3o–x, *nodal* and *lefty* is broadly expressed in the lateral ectoderm, even after halved embryos form a sphere. Given that FoxQ2 selectively represses the auto-regulatory loop of *nodal* but does not block its initial expression[21], we

hypothesize that the Nodal/FoxQ2-free region is formed by the positional shift of the *foxQ2*-expressing anterior end in response to posterior signals. This shift re-establishes the morphogen gradient along the D-V axis, starting from the Nodal/FoxQ2-free region. Support for this model comes from the observation that BMP2/4 becomes active after sphere formation (Supplementary Fig. 10). Based on these observations, our findings suggest that the re-establishment of the anterior–posterior (A–P) axis acts as an indirect trigger for the formation of the dorsal–ventral (D–V) axis. While the A–P axis does not directly instruct D–V specification, its proper reconstruction appears to create a necessary permissive condition for the initiation of D–V polarity. One possible mechanistic explanation for this is based on previous studies showing that the D–V axis can only be established once the Nodal repressor, FoxQ2, is cleared from the ectoderm[21]. During the ambiguous period of A–P axis re-formation, FoxQ2 expression is transiently detected and then disappears. Although FoxQ2 does not prevent the initial activation of *nodal*, its presence is thought to interfere with the autoregulatory loop required for sustained Nodal expression[21]. Therefore, it is reasonable to assume that the autoregulatory loop of Nodal—required for robust D–V axis formation[26]—can only be stably activated after the anterior shift and clearance of FoxQ2 have been completed. In this way, A–P axis re-formation indirectly triggers D–V axis establishment by ensuring a cellular environment conducive to stable Nodal expression. A similar phenomenon has been observed in *Xenopus laevis*, where dorsal and ventral ends are juxtaposed during wound healing, and both ends shift from their original positions when blastulae are bisected[39]. Since β-catenin is stabilized at the dorsal end by cortical rotation in Xenopus, inducing a morphogen gradient along the D-V axis[40], it is possible that a similar mechanism triggers self-organization in sea urchins. In sea urchins, β-catenin is stabilized at the vegetal pole, and the adjacency of the anterior and posterior ends initiates self-organization. Indeed, stable A-P axis formation is important for D-V axis formation, since A-P

axis deficient halved embryos, in which Δcadherin was injected into, are covered with long and immotile cilia (Supplementary Fig. 13) as in reported Δcad embryos, indicating that they are composed only of anterior neuroectoderm[20]. The reason for the broad *nodal* and *lefty* expression during the cup and sphere stages is unclear and should be addressed in future studies. One report suggest that redox state regulate the D-V axis formation[41]. Live imaging of mitochondria distribution in halved embryos may provide further insights into this process.

Based on the molecular mechanisms of self-organization revealed in this study, we define self-organization as the process in which axes and molecular gradient information is reorganized within a newly defined individual boundary, without de-differentiation, in response to signals from the organizer. This study not only demonstrates the first experimental proof of axes re-formation during self-organization but also highlights the robustness and flexibility of embryonic developmental programs. The fact that sea urchin embryos, and potentially other species, can autonomously re-establish their developmental axes and molecular gradients in response to disruptions suggests that these systems are hardwired for resilience. This intrinsic ability to re-form body axes without external cues points to the evolutionary significance of self-organization, likely as a safeguard against environmental or developmental perturbations. Understanding these molecular mechanisms provides new insights into how early embryonic systems maintain developmental fidelity. Moreover, the parallels between sea urchin and Xenopus embryos hint at conserved evolutionary pathways regulating self-organization, possibly extending to other deuterostomes. In a broader context, our findings may have implications for regenerative biology and artificial tissue engineering, where inducing or harnessing similar self-organizing principles could be critical for tissue repair and organ regeneration. The discovery of species-specific features, such as the cell elongation observed in *H. pulcherrimus*, further suggests that there may be as-yet-undiscovered diversity in self-organization mechanisms across species. Investigating these variation and/or similarity in depth could open new avenues for understanding developmental plasticity and robustness, providing valuable insights into both evolutionary biology and biomedical applications.

## Methods

### Animal collection and embryo culture

Adults of *Hemicentrotus pulcherrimus* and *Temnopleurus reevesii* were collected around Shimoda Marine Research Center (University of Tsukuba) and Marine and Coastal Research Center (Ochanomizu University) under the special harvest permission of prefectures and the Japan Fishery cooperative. *Strongylocentrotus intermedius* were gifted for experimental purposes from Chirippu Fishery Cooperative, and the experiment using *Strongylocentrotus purpuratus* was supervised by Prof. Gary Wessel (Brown University, Providence, RI, USA). Gametes were collected by injecting 0.5 M KCl into the body cavity. The embryos were cultured in plastic dishes or glass beakers filled with filtered natural seawater (FSW) containing 50 μg/ml of kanamycin sulfate (Nacalai Tesque Inc., Kyoto, Japan) and kept at 15 °C until an appropriate stage. FM1-43 FX (ThermoFisher Scientific, Waltham, MA, USA) was treated with a final 4 μM in FSW from 1 h before observation. Cytochalasin D (FUJIFILM Wako Pure Chemical Co., Osaka, Japan), (-)-blebbistatin (FUJIFILM), cyclopamine (FUJIFILM), and 4-aminopyridine (FUJIFILM) were treated with 2 μM, 30 μM, 6.25 μm, and 2 mM, respectively, from the flat or cup shape stage. Actinomycin D (FUJIFILM) and cycloheximide (FUJIFILM) were treated with 25 μg/μl and 2 mM from 20 min after fertilization. U0126 (Sigma-Aldrich, St Louis, MO, USA) and Y27632 (Abcam, Cambridge, UK) were added with 10 μM and 1 mM from 2-cell stage. C59 (Funakoshi Co., Ltd., Tokyo, Japan) was treated from the cup stage at 0.8 mM. SP600126 (FUJIFILM) was treated from the sphere shape stage at 2 μM. All reagents were

diluted with FSW, and the same volume of dimethyl sulfoxide (DMSO) was used as a negative control.

### Counting cell number

For cell counting in embryos, the embryos were fixed with 3.7% formaldehyde in FSW for 10 minutes and washed several times with PBS-T ([PBS] Nippon Gene, Tokyo, Japan 0.1% Tween-20). Nuclei were then stained with DAPI. The embryos were observed under a confocal microscope, and the number of cells was determined. To avoid miscounting, nuclear signals were marked digitally during analysis.

### Whole-mount in situ hybridization and immunohistochemistry

All gene information is available at HpBase (Genome & Transcriptome database for *H. pulcherrimus*[8]). Tetraspanin (tspn: HPU_14020) and ZO-1 (HPU_12399) of *H. pulcherrimus* were cloned with primers written in below.

Tspn-F:
5′-CATCGATTCGAATTC]ATGGGTATGGATCTGGGAGGGTGCGC-TAAG-3′

Tspn-R:
5′-TTCTAGAGGCTCGAG]CTAGACAACGTCCTCTCCCTTGGA-GATGCC-3′

ZO-1-F:
5′-CATCGATTCGAATTC]ATTAAACCAGGTAGCCCTGCA-GAAAGCTCT-3′

ZO-1-R:
5′-TTCTAGAGGCTCGAG]CTGTCCTCTGA-TATCCTTGACCTGGGAGCG-3′

Forwerd primers have EcoRI sequence, and Reverse primer have XhoI sequence for inserting PCR products to pCS[2+] vector with infusion cloning kit (TAKARA BIO INC., Shiga, Japan).

Whole-mount in situ hybridization was performed as previously described[42]. Embryos were fixed with 3.7% formaldehyde in FSW for overnight at 4 °C. Fixed embryos were washed with MOPS buffer (0.1 M MOPS, pH 7.0, 0.5 M NaCl, 0.1% Tween-20) for 7 min x 7 times, then pre-hybridized with hybridization buffer (70% formamide, 0.1 M MOPS, pH 7.0, 0.5 M NaCl, 0.1% Tween-20, 1% BSA) at 50 °C for 1 h. Samples were hybridized with 0.4 ng/μl Digoxygenin (Dig)-labeled RNA probes in hybridization buffer at 50 °C for 5–7 days. Subsequently, samples were washed with MOPS buffer 7 min x 7 times and blocked with 2.5% skim milk (Cell Signaling Technology, Danvers, MA, USA) in MOPS buffer at room temperature (R.T) for 1 h. Specimens were incubated for overnight at 4 °C with 1/1500 anti-Dig antibody conjugated with alkaline phosphatase (AP) (Roche Holding AG, Basel, Switzerland) in MaxBlot Solution 2 (MBL, Tokyo, Japan). Samples were washed with MOPS buffer, and MOPS buffer were substituted with AP buffer (0.1 M Tris, pH 9.5, 100 mM NaCl, 50 mM MgCl$_2$, 0.1% Tween-20, 1 mM Levamisole) by 2 times for 7 min each. Dig was detected by the reaction of AP with Staining solution (10% dimethylformamide, 3.15 μl/ml NBT and 2.45 μl/ml BCIP [Promega Corporation, Madison, WI, USA] diluted in AP buffer) in the dark until the signal was visualized under the microscope. Whole-mount in situ hybridization chain reaction was performed as described in previous study[43] with small modification.

Immunohistochemistry was also performed as previously described[42]. For immunohistochemistry, embryos were fixed with 3.7% formaldehyde in FSW for 10 min at R.T. For detecting phosphorylated-Smad2/3 (pSmad2/3), embryos were fixed with 4% paraformaldehyde in FSW for 10 min. Fixed embryos were washed with PBS-T 7 min x5 times and blocked with 5% lamb serum (ThermoFisher Scientific) in PBS-T for 1 h. Specimens were incubated for overnight at 4 °C with each antibody in MaxBlot Solution 1 (MBL). Anti-FoxQ2 (in-house made[44]), anti-pSmad2/3 (Abcam) and anti-pSmad1/5/8 (Cell Signaling Technology) antibody was diluted to 1/100, 1/100 and 1/1000 in Max-Blot solution 1, respectively. Samples were washed with PBS-T buffer

and reacted with 1/2000 diluted secondary antibody (ThermoFisher Scientific) in MaxBlot Solution 2 for 2 h at R.T in dark.

## Microinjection

Micro-injection into fertilized eggs and blastomeres was performed as previously described[45]. Each morpholino anti-sense oligo nucleotides (MO) and mRNA for micro-injection were dissolved in the injection buffer (24% glycerol, 20 mM HEPES pH 8.0, and 120 mM KCl). The concentration in needle and the sequence of morpholino and mRNA were as follows:

BMP2/4-MO (400 µM): 5′- GACCCCAATGTGAGGTGGTAACCAT -3′ (Specificity was previously confirmed[44]);

FoxQ2-MO (200 µM): 5′- TCATGATGAAATGTTGGAACGAGAG -3′ (Specificity was previously confirmed[42]);

Nodal-MO (200 µM): 5′- AGATCCGATGAACGATGCATGGTTA -3′ (Specificity was previously confirmed[46]);

Lefty-MO (400 µM): 5′- AGCACCGAGTGATAATTCCATATTG -3′ (Specificity was previously confirmed[42]);

Tetraspanin-MO1 (3.8 mM): 5′- GATCCATACCCATCTTGAATTA-TAT -3′

Tetraspanin-MO2 (200 µM): 5′- CTTGACTACACGCTTCGTAAA-CAAA -3′

ZO-1-MO1 (3.8 mM): 5′- CATATCCGACATCCATCATGG -3′

ZO-1-MO2 (1.0 mM): 5′- TCTCTGTCCACCTCTTTATAA -3′

Zebrafish UROD-MO (200 µM): 5′- GAATGAAACTGTCCTTATC-CATCA -3′,

Δcadherine mRNA (400 ng/µl), Lifeact mcherry mRNA (500 ng/µl), ß-catenin-Venus mRNA (500 ng/µl) and membrane TagRFP (100 ng/µl).

## Micromanipulation

Embryos were fertilized in the FSW containing 1 mM 3-amino-1,2,4-triazole (ATA) to prevent the fertilization envelope from hardened[47]. Fertilized eggs were cultured at 15 °C until 2-cell stage and removed fertilization envelope by pipetting. Ease for manipulation, the embryos were treated with $Ca^{2+}$-free seawater (CFSW; 435 mM NaCl, 9.3 mM KCl, 24.5 mM $MgCl_2$, 25.5 mM $MgSO_4$, 2.15 mM $NaHCO_3$, Woods Hole, MBL). Fire-burned and pulled thin-wall glass capillary (Sutter Instrument, Novato, CA, USA) was used for cutting embryos. The protocol for making a manipulation needle was previously described[45]. All plastic dishes and glass capillaries for cutting embryos were coated with 3% BSA or lamb serum. The cut embryos were maintained in the small holes on the 1% agarose gel in FSW, which were made by the heated tweezers.

## Total RNA sequencing and analysis

Total RNA was collected from 100 control embryos and 200 halved embryos with RNeasy Plus Mini Kit (Qiagen) at each developmental stage. For total RNA sequencing, we utilized the Macrogen Total RNA sequencing service.

Data retrieval and pre-processing: The reference genome file (HpulGenome_v1_scaffold.fa) and the annotation files (HpulGenome_v1.gff3 and HpulGenome_v1_annot.xlsx) of *H. pulcherrimus* were retrieved from HpBase. We performed Trimmomatic (v0.39) against 54 of FASTQ files and trimmed Illumina adapter sequences with our original list (https://bioinformatics.riken.jp/ramdaq/ramdaq_annotation/mouse/all_sequencing_WTA_adopters.fa).

Mapping to the reference genome and quantification of gene expression: The index file of the STAR aligner (v2.7.10a) was generated with the *--runmode genomeGenerate* and then the trimmed FASTQ files were mapped to the reference genome using STAR with the default parameters. FeatureCounts (v2.0.1) was used to quantify the gene expression levels of each sample. The gene expression matrix was generated by counting the number of reads mapped to each gene.

Data quality control (QC): FastQC (v0.11.9) was used to check the quality of raw and trimmed reads. MultiQC (v1.12) was used to summarize the results of FastQC. Using R (v4.4.2), we generated a heatmap of the correlation between samples using the R package ggplot2 (v3.5.1). prcomp is used to perform PCA and the mapping rate and the number of detected genes were evaluated on the score plot.:Differentially expressed gene (DEG) detection:

edgeR (v4.4.1) and DESeq2 (v1.46.0) were used to detect DEGs. The DEGs were visualized using the R package ggplot2 (v3.5.1).

Reproducibility: All the data analyses were performed on a workflow using Snakemake (v6.5.3) (https://github.com/kokitsuyuzaki/urchin-workflow). All the tools are pre-installed in the following three Docker containers, and executed via Singularity (v3.5.3), so that data analyses are reproducible in any computational environment.

- koki/urchin_workflow_bioconda:20220527: Trimmomatic, FastQC, STAR, FeatureCounts
- quay.io/biocontainers/multiqc:1.12--pyhdfd78af_0: MultiQC
- koki/urchin_workflow_r:20250116: Dimensionality reduction, DEGs analysis, visualization

## Cell labeling with DiI

CellTrack™ CM-DiI (DiI) (Thermo Fisher Scientific) was dissolved into 100% EtOH to final 1 µg/µl concentration. Diluted DiI was loaded into the glass capillary, which is used for micro-injection. DiI that flows into the seawater becomes a small crystal at the tip of the capillary, since DiI is not dissolved in water/seawater. DiI crystal was pressed to the target portion of halved embryos on poly-L-lysin (Sigma-Aldrich) coated plastic dish for 2 min. After the stained signal is confirmed by a fluorescent microscope (IX70 [Olympus, Tokyo, Japan]), the labeled embryos were kept at 15 °C until the appropriate stages.

## Statistical analysis

No statistical methods were used to predetermine the sample size. All n numbers are described in figure. To compare the two groups in Fig. 2a (median of control = 98.5, median of half x2 = 97, W statistic = 532.5, *p*-value = 0.8877), Fig. 2b (median of control = 397, median of half x2 = 378, W statistic = 820.5, *p*-value = 0.08695), we used Wilcoxon rank-sum test with a significance level of 0.05. To compare the four groups in Fig. 2k, we used one-way ANOVA followed by Tukey's Honestly Significant Difference (HSD) test with a significant level of 0.01 or 0.05, and following *F* value (F) and degrees of freedom [d.f.]. For Fig. 2k: F = 72.406, d.f. = 3.

## Reporting summary

Further information on research design is available in the Nature Portfolio Reporting Summary linked to this article.

## Data availability

Sequence data can be found in the genome database of *Hemicentrotus pulcherrimus*, HpBase (http://cell-innovation.nig.ac.jp/Hpul/). RNA-seq data has been deposited at the NCBI GEO database under accession code GSE289178. Source data are provided with this paper.

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

## Acknowledgements

We thank Dr. Brad Shuster for the essential reagent. We thank Dr. Jenifer Croce and Dr. Guy Lhomond for essential technical advice. We thank Prof. M. Kiyomoto, Prof. D. Shibata, M. Ooue, J. Takano, Y. Uchida, G. Northen, H. Abe, M. Washio, M. Yamaguchi, and JF Izu/Shimoda for collecting and keeping the adult sea urchins. We thank Prof. Gary Wessel for crucial cooperation for experiment. This work is supported by JST PRESTO Grant number JPMJPR194C, JST A-STEP Grant number JPMJTR204E, JSPS KAKENHI Grant number 23K23933 and 24K21959 to S.Y., and JST PRESTO Grant number JPMJPR1945 and JSPS KAKENHI Grant Numbers 19K20406 and 23K11312 to K.T.

## Author contributions

Studies were designed by H.S. and S.Y. Data were acquired by H.S., J.Y., K.T., and S.Y. The manuscript was written by H.S. and S.Y. with input from all authors.

## Competing interests

The authors declare no competing interests.
