## [Peer Review file · Nature Communications]

Unraveling the regulative development and molecular mechanisms of identical sea urchin twins

Corresponding Author: Professor Shunsuke Yaguchi

Version 0:

Reviewer comments:

Reviewer #1

(Remarks to the Author)

In this manuscript Yaguchi and coll. address a key and long standing question in developmental biology: How do the early blastomeres regulate and reform a harmoniously patterned larva when they are isolated at an early stage. Since the experiment of Driesch, showing that the first four blastomeres can regulate and reform a normal pluteus larva, this question has remained unanswered in the field.

This study, which to my knowledge is the first in the sea urchin is therefore particularly welcome. In the first part of the manuscript the authors report that halved embryos, when bisected at the 2-cell stage undergo a striking and unexpected sequence of morphogenesis going from a flat shape to a cup shape and finally to a spherical shape. Intrigued by this unusual observation, they attempt to understand how the embryoid reorganized and how it went from a flat to a spherical morphology. They then demonstrate in a beautiful study of cell biology that the actomyosin and septate junctions provide the driving force for the shape transitions of halved embryos. Using inhibitors of actin polymerisation they show that this sequence of morphogenesis does not occur if actin polymerisation is inhibited. Similarly they show that the septate junction associated gene Tetraspanin is required for the embryoids to reach a spherical stage. From these experiments they conclude that the morphogenetic sequences of the embryoids does not involve the signaling pathways used for axis specification but that it relies instead on cell autonomous processes involving actin remodeling and septate junctions. Then, they go on and focus on the question of how are the A/P and D/V axes of the embryoids reorganized. From the analysis of the expression of a small number of anterior and posterior genes, they conclude that the A/P axis collapses in halved embryoids and recovers a few hours later but in a perpendicular orientation and they also found that blocking the beta catenin pathway prevents this reorganization of the A/P axis. Regarding the D/V axis the authors observed a complex sequence of expression of nodal and lefty with nodal being asymmetrically expressed in the flat embryoids, ubiquitous expression in the cup stage and again asymmetrically expressed 4h after the sphere stage.

The experiments reported in this paper are beautifully executed paper and I have only a few questions/remarks.

1. In the discussion section of the paper, the authors argue that the D/V axis is triggered by the reformation of the A/P axis but the arguments put forward are not completely convincing.

2. Can the D/V axis be reformed in the absence of the A/P axis? (and of beta catenin signaling?)

3. What could be the reasons for the broad expression of nodal and lefty in halved embryos? and why does Lefty disappear at the sphere stage (Fig. 3w) if nodal is strongly and ubiquitously expressed at this stage? have the authors considered the possibility that this ectopic activation of pSmad2 could result from perturbation of a Nodal antagonist? There are many players that contribute to the regulation of nodal expression including redox gradients. Have the authors considered the possibility that redox gradients could be perturbed in the halved embryos?

4. I have the same question regarding pSmad2/3 (Fig S7). How is Lefty expressed in these embryos? is the reaction diffusion mechanism operating in these embryos?

5. Do the authors know which BMP ligand is responsible for the pSmad1 signals in halved embryos?

6. Regarding the reformation of the D/V axis: is there any evidence that the new D/V axis is reformed by shuttling of BMP4 and do you know if chordin is expressed in a restricted manner in these embryoids? Would it be possible to double stain the embryoids for pSmad2/3 and pSmad1?

7. the authors argue that axis re-formation is hardwired in the genome and that "the intrinsic ability to repattern body axes

without external cues points to the evolutionary significance of self organization"; I think that the body axes are not "repatterned" but that they are reformed. This is partially similar to what happens during regeneration, when novel organs have to be reconstructed from very little spatial cues.

Clearly, deciphering the molecular mechanisms underlying development of the embryoids will require additional studies but the findings reported in this first study are greatly valuable and the description of the morphogenetic sequence of halved embryos is very interesting. Indeed, we would have liked to know more about the scaling of halved embryos and what happens to the embryoids in terms of size regulation. Genes involved in size regulation and scaling have been characterized in *Xenopus* or in the fly. It would have been very interesting to study these genes in the context of this study but I understand that this is probably beyond the scope of this first paper.

(there are I think several tetraspanin genes in the sea urchin genome. Which one are you referring to?)

Reviewer #2

(Remarks to the Author)

Suzuki and co-authors provide a detailed analysis of the development of sea urchin twin embryos, derived by separation of blastomeres at the 2-cell stage. This is the first analysis of its kind: even though the phenomenon has been well known for over a century, the morphogenetic and molecular phenomena allowing the development of fully formed embryos from isolated blastomeres has been the subject of very little investigation. With a careful examination of the process, the authors make a significant contribution to the field of cell biology and developmental biology, uncovering unexpected ways in which isolated blastomeres form a blastula and establish their antero-posterior axis. The findings are of high interest, although several points of concern warrant further clarification:

Major concern:

1. The authors observe a "flat" phenotype, in which the daughter cells of isolated blastomeres form a sheet, which then curves into a cup and finally closes a sphere. This is an unexpected phenotype, as sea urchin cells are normally forming a compact cluster. There are several aspects of the morphogenetic events underlying this transition that need to be clarified:

a. Are the authors absolutely sure that the flat phenotype is not due to adhesion of the sea urchin cells to the substrate they are cultured in?

b. The authors state that cell division is not affected, as time matched half-embryos have half the number of cells compared to control whole embryos. However, the method used to count the cells is not specified. Without that information it is impossible to gauge the accuracy of this measurement. Not just the number of cell divisions, but also their orientation may be altered.

c. The morphogenetic transition from flat, to cup and then to sphere is depicted as due to elongation of mostly static cells. However, the supplemental movie provided shows quite a bit of movement of the whole tissue and one can discern some migratory cells (PGCs perhaps?) and some cells that are left out of the sphere. This suggests that there may be cellular rearrangements occurring during this process, which are not normally observed during development of intact embryos. If this were the case, it would have implications also for the hypothesis put forward regarding the mechanisms of antero-posterior axis formation.

The points above could be clarified if the authors analyzed their confocal time-lapses of half embryos to extract cell division events, cell rearrangements and cell death. Alternatively other ad hoc experiments could be devised to address each point.

2. The morphogenetic events underlying the transition between flat, cup and sphere are unclear.

a. The authors show that the transition to cup is accompanied by excessive elongation of cells, which are transiently longer than cells in whole blastulae. However, the comparison is made only to one blastulation end-point for whole embryos. What do whole embryo cells look like at earlier stages? Can the authors compare with time-matched whole embryos?

b. The authors show an accumulation of actin at the basal side of halved embryos. However the images also show actin signal at the apical side. How much stronger is the signal on the basal side? Is the anisotropy in actin localization similar or stronger than in whole embryos? In other words, is this accumulation a "normal" thing that sea urchin cells do, and that brings about the cupping in halved embryos, or is this a response to the lack of cell-cell contacts on the basal side? The authors suggest it is the former, as they state in the discussion that basal actin accumulation is necessary for blastula formation, but do not provide evidence or a citation for this statement.

c. Is it always the basal side that bends? Do the authors ever observe "inside-out blastulae"?

d. Depolymerization of actin with cytochalasin D affects all actin in the cell, so the authors cannot state that the basal actin accumulation is necessary for the process. They would have to disrupt only the basal actin to make that statement.

e. Cell division is probably also blocked by treatment with cytochalasin D: can the authors exclude that cell division is necessary for cupping?

d. The tetraspanin morpholino experiment is quite striking, although several controls are missing. Is cell division impaired? Is apoptosis induced? Do these embryos produce a normal hyaline layer?

3. Stronger evidence is needed to support the proposed mechanism for anteroposterior axis formation. The authors bring forward a model in which flat embryos have an anteroposterior axis, equivalent to the one that would have formed if the blastomere had not been isolated, that gets re-organized upon closure of the cup into a sphere. They suggest that the original anterior-most and posterior-most ends of the half embryo meet once the cup is closed and that the axis is reset at that point, due to a reactivation of Wnt/b-catenin signalling. The evidence provided in support of this model is enticing, but not quite compelling. In particular,

a. To show that the anterior-most and posterior-most cells of the half-embryo meet when the cup closes, the authors use HCR for anterior and posterior genes. While they beautifully show the domains coming closer and then separating again after closure of the sphere, this technique does not allow distinguishing between changes in gene expression and cell movement. It is therefore unclear whether cells that originally expressed anterior and posterior genes come into contact and then change their expression profile, or if they move, before or after closure of the cup. Also, the HCR on fixed samples at the sphere stage do not provide information on the position of the FoxQ2 and Ax11 expression domains with respect to the point of cup closure. To clarify this point the authors could perform a double tracing experiment, marking one anterior and one posterior cell with dyes of different color and then follow their relative position at cup, early sphere, late sphere and gastrula. Images for the same half embryos could be taken at multiple time points. Alternatively, they could take time lapses of halved embryos and track anterior and posterior cells across flat, cup and sphere stages.

b. The authors show that nuclear b-catenin is present in halved embryos, and state that it is not normally active in whole embryos at equivalent stages. However, domains of nuclear b-catenin have been described at blastula stages in other urchins (see Logan, 1999) and whether the sphere stage corresponds to blastula stages in which some cells may still have nuclear b-catenin in whole embryos is unclear. The authors should provide similar data on b-cat localization for time matched whole embryos.

c. The authors state that nuclear b-catenin signal is observed in a domain adjacent to the original posterior cells: how was this determined?

d. To experimentally prove the need for Wnt/b-catenin signalling to re-organize the axis, the authors analyze halved embryos injected with $\Delta Ecad$. The experiment is not well described, but we assume that the zygotes were injected and then blastomeres were separated at 2-cell stage. In this case, all Wnt/b-catenin signalling would be impaired, meaning these embryos would have failed to generate any axis, and not only to re-organize it upon closure. A more convincing experiment would be blocking Wnt/b-catenin signalling after the primary axis has been established, which could be achieved using inhibitors like C59. This should result in embryos that have anteriorly fated cells (expressing FoxQ2 or making an apical tuft) right next to posterior invaginating cells.

e. If the contact between anterior and posterior domains in the sphere stage is necessary to trigger axis reorganization, the axis should stay the same if the contact were to be prevented. Can the author keep the halved embryos flat and check whether the axis remains as initially set or re-organized?

Minor concern:

1. In the discussion the authors argue that the mode of cup and sphere formation is passive, as opposed to an active boundary recognition mode. What is meant with these two terms, passive and active, is very unclear. Cells elongate and curve the tissue to close it up, which does not seem a passive mechanism.

2. In the discussion, several parallels are drawn between dissociated sea urchin and sea star cells, which ignore the fact that sea star embryos do not have a hyaline layer and become quite flattened even just after removal of their fertilization envelope: it is not surprising that sea star cells would not form a compact cluster upon blastomere separation. The reasons why sea star blastomeres form a flat sheet upon dissociation may be entirely different from the reasons why sea urchin cells do the same.

Reviewer #4

(Remarks to the Author)

Version 1:

Reviewer comments:

Reviewer #1

(Remarks to the Author)

The authors have answered most of the questions and concerns I had. The only point that I think remains unclear is when the authors discuss about the role of FoxQ2. They argue that the completion of FoxQ2 expression to the anterior end is essential for initiating D/V axis formation. However, FoxQ2 function is not required for D/V axis formation in the unperturbed embryo and nodal expression is normal in the FoxQ2 morphants. Wouldn't we expect nodal to be expressed prematurely in the absence of this repressor of nodal expression? There is something that I don't get here. Otherwise, they have clarified all the other points that were too vague and they have provided numerous supplementary data and additional analyses that further increase the interest of the paper.

Reviewer #2

(Remarks to the Author)

The authors have addressed all the points I raised. I congratulate the authors on this thorough and elegant work!

Reviewer #1 (Remarks to the Author)

In this manuscript Yaguchi and coll. address a key and long standing question in developmental biology: How do the early blastomeres regulate and reform a harmoniously patterned larva when they are isolated at an early stage. Since the experiment of Driesch, showing that the first four blastomeres can regulate and reform a normal pluteus larva, this question has remained unanswered in the field.

This study, which to my knowledge is the first in the sea urchin is therefore particularly welcome. In the first part of the manuscript the authors report that halved embryos, when bisected at the 2-cell stage undergo a striking and unexpected sequence of morphogenesis going from a flat shape to a cup shape and finally to a spherical shape. Intrigued by this unusual observation, they attempt to understand how the embryoid reorganized and how it went from a flat to a spherical morphology. They then demonstrate in a beautiful study of cell biology that the actomyosin and septate junctions provide the driving force for the shape transitions of halved embryos. Using inhibitors of actin polymerisation they show that this sequence of morphogenesis does not occur if actin polymerisation is inhibited. Similarly they show that the septate junction associated gene Tetraspanin is required for the embryoids to reach a spherical stage. From these experiments they conclude that the morphogenetic sequences of the embryoids does not involve the signaling pathways used for axis specification but that it relies instead on cell autonomous processes involving actin remodeling and septate junctions. Then, they go on and focus on the question of how are the A/P and D/V axes of the embryoids reorganized. From the analysis of the expression of a small number of anterior and posterior genes, they conclude that the A/P axis collapses in halved embryoids and recovers a few hours later but in a perpendicular orientation and they also found that blocking the beta catenin pathway prevents this reorganization of the A/P axis. Regarding the D/V axis the authors observed a complex sequence of expression of nodal and lefty with nodal being asymmetrically expressed in the flat embryoids, ubiquitous expression in the cup stage and again asymmetrically expressed 4h after the sphere stage.

The experiments reported in this paper are beautifully executed paper and I have only a few questions/remarks.

1. In the discussion section of the paper, the authors argue that the D/V axis is triggered by the reformation of the A/P axis but the arguments put forward are not completely convincing.

Thank you for your comment. Our conclusion is based not only on the present observations but also on previous studies. In fact, the establishment of the A–P axis—specifically, the completion of FoxQ2 localization to the anterior end—is essential for initiating the D–V axis even in normal embryos. We intended to emphasize that a

similar mechanism likely operates in the half-embryos, where FoxQ2 expression gradually shifts over time. To better convey this idea, we have revised the Discussion section to clarify and elaborate on this point. “Based on these observations, our findings suggest that the re-establishment of the anterior–posterior (A–P) axis acts as an indirect trigger for the formation of the dorsal–ventral (D–V) axis. While the A–P axis does not directly instruct D–V specification, its proper reconstruction appears to create a necessary permissive condition for the initiation of D–V polarity. One possible mechanistic explanation for this is based on previous studies showing that the D–V axis can only be established once the Nodal repressor, FoxQ2, is cleared from the ectoderm. During the ambiguous period of A–P axis re-formation, FoxQ2 expression is transiently detected and then disappears. This dynamic pattern likely interferes with the stable expression of Nodal, which is essential for initiating the D–V axis. Therefore, it is reasonable to assume that the autoregulatory loop of Nodal—required for robust D–V axis formation—can only be stably activated after the anterior shift and clearance of FoxQ2 have been completed. In this way, A–P axis re-formation indirectly triggers D–V axis establishment by ensuring a cellular environment conducive to stable Nodal expression.”

2. Can the D/V axis be reformed in the absence of the A/P axis? (and of beta catenin signaling?)

No, the D/V axis will not be reformed in A/P-absent half-embryos (actually in normal embryos, too). We showed this in Supplementary Fig. 4a and 13, Δ cad mRNA injected embryos.

3. What could be the reasons for the broad expression of nodal and lefty in halved embryos? and why does Lefty disappear at the sphere stage (Fig. 3w) if nodal is strongly and ubiquitously expressed at this stage?

Thank you for your comment. Unfortunately, this phenomenon remains unresolved even within our own analyses, and we are currently unable to provide a definitive answer. The timing and location of the initial expression of *nodal* and *lefty* have not yet been fully determined even in normal embryos, and the papers from the original group that reported *nodal* expression have also presented differing results over time (Duboc et al. 2004 Developmental Cell; Haillet et al. 2015 PLOS Biology). This suggests that *nodal* expression is inherently unstable, likely due to the delicate balance between Nodal-mediated auto-activation, Nodal-induced *lefty* expression, and Lefty-mediated repression of Nodal. In addition, in halved embryos, the FoxQ2 domain—a known repressor of *nodal* expression—continues to shift within the ectoderm between the cup and sphere stages, which may further contribute to the instability of *nodal* expression. We have now added a brief note about this unresolved issue to the text of Fig. 3w (Pg. 13, Ln. 289).

Have the authors considered the possibility that the ectopic activation of pSmad2 could result from perturbation of a Nodal antagonist ?

Yes, this is somewhat related to the discussion above. We speculate that the transient expansion of pSmad2/3 throughout the ectoderm was caused by temporary changes in Lefty (a Nodal antagonist) activity.

There are many players that contribute to the regulation of nodal expression including redox gradients. Have the authors considered the possibility that redox gradients could be perturbed in the halved embryos?

Thank you for this thoughtful comment. We added the sentence to discussion part about the redox. “The reason for the broad *nodal* and *lefty* expression during the cup and sphere stages is unclear and should be addressed in future studies. One report suggest that redox state regulate the D-V axis formation³⁹. Live imaging of mitochondria distribution in halved embryos may provide further insights into this process.”

4. I have the same question regarding pSmad2/3 (Fig S7). How is Lefty expressed in these embryos? is the reaction diffusion mechanism operating in these embryos?

As discussed above, the precise expression patterns and functions of Lefty remain unclear. We plan to investigate the reaction–diffusion dynamics of Nodal (as reflected by pSmad2/3) and Lefty in greater detail in future studies.

5. Do the authors know which BMP ligand is responsible for the pSmad1 signals in halved embryos?

Thank you for this comment. Based on the history of studies on D–V axis formation in sea urchins (e.g. Duboc et al. 2004 Developmental Cell), as well as our current analysis of nodal expression from the cup to sphere stages, we assume that BMP2/4 acts upstream of pSmad1/5/8 in this context. In particular, previous work (Yaguchi et al., 2011, Development) has shown that pSmad1/5/8 signaling is not detected until BMP2/4 activity becomes evident. Therefore, it is reasonable to consider BMP2/4 as the upstream regulator of pSmad1/5/8 signaling in the current study as well.

6. Regarding the reformation of the D/V axis: is there any evidence that the new D/V axis is reformed by shuttling of BMP4 and do you know if chordin is expressed in a restricted manner in these embryoids?

Thank you for this comment. We had not previously examined chordin expression in this context, so we conducted new experiments to assess it. Our results show that in halved embryos, *chordin* exhibits asymmetrical expression similar to that in control embryos (a figure has been added). Previous studies have shown that Chordin, like other D–V axis–related TGF- β family members, is expressed downstream of Nodal during early development (e.g. Lapraz et al. 2009 PLoS Biol), and is therefore assumed to be localized to the future ventral side. In addition, as BMP2/4 has been reported to act within the same signaling pathway, we infer that it is likely subject to the same regulatory logic. While we were unable to directly observe the shuttling mechanism suggested by the reviewer, we consider it plausible that BMP2/4 is redistributed toward the future dorsal side, as previously proposed in control embryos (Yaguchi et al. 2011 Development).

Would it be possible to double stain the embryoids for pSmad2/3 and pSmad1?

It is impossible because both anti-pSmad2/3 and anti-pSmad1/5/8 were raised from rabbits. We would really like to do this experiment in future.

7. the authors argue that axis re-formation is hardwired in the genome and that "the intrinsic ability to repattern body axes without external cues points to the evolutionary significance of self organization"; I think that the body axes are not "repatterned" but that they are reformed. This is partially similar to what happens during regeneration, when novel organs have to be reconstructed from very little spatial cues.

We agree with this comment, the body axes in halved embryos are “re-formed”, not re-pattern. Disturbed A-P and D-V axis are re-formed with intrinsic pathways. We fixed the sentence.

(there are I think several tetraspanin genes in the sea urchin genome. Which one are you referring to?)

We referred to HPU_14020 (Hp-Ttrspn_16-like), which is homologous to Sp-tetraspanin (LOC751836 in Echinobase: <https://www.echinobase.org/echinobase/>), previously used in Jonusaite et al. (<https://doi.org/10.1016/j.ydbio.2022.12.007>). The HPU number is referred to in the Methods section, along with ZO-1 (HPU_12399).

*Clearly, deciphering the molecular mechanisms underlying development of the embryoids will require additional studies but the findings reported in this first study are greatly valuable and the description of the morphogenetic sequence of halved embryos is very interesting. Indeed, we would have liked to know more about the scaling of halved embryos and what happens to the embryoids in terms of size regulation. Genes involved in size regulation and scaling have been characterized in *Xenopus* or in the fly. It would have been very interesting to study these genes in the context of this study but I understand that this is probably beyond the scope of this first paper.*

Thank you for your thoughtful and insightful comments. We fully agree with the reviewer and are committed to continuing this line of research to make significant advances in the field.

Reviewer #2 (Remarks to the Author)

Suzuki and co-authors provide a detailed analysis of the development of sea urchin twin embryos, derived by separation of blastomeres at the 2-cell stage. This is the first analysis of its kind: even though the phenomenon has been well known for over a century, the morphogenetic and molecular phenomena allowing the development of fully formed embryos from isolated blastomeres has been the subject of very little investigation. With a careful examination of the process, the authors make a significant contribution to the field of cell biology and developmental biology, uncovering unexpected ways in which isolated blastomeres form a blastula and establish their antero-posterior axis. The findings are of high interest, although several points of concern warrant further clarification:

Major concern:

1. The authors observe a “flat” phenotype, in which the daughter cells of isolated blastomeres form a sheet, which then curves into a cup and finally closes a sphere. This is an unexpected phenotype, as sea urchin cells are normally forming a compact cluster. There are several aspects of the morphogenetic events underlying this transition that need to be clarified:

a. Are the authors absolutely sure that the flat phenotype is not due to adhesion of the sea urchin cells to the substrate they are cultured in?

Yes, to prevent the embryos from adhering to the substrate, we cultured them in wells made in agarose (as described in the Methods section). In addition, we performed the same experiment using BSA-coated dishes, and the embryos exhibited the same phenotype without adhering to the dish.

b. The authors state that cell division is not affected, as time matched half-embryos have half the number of cells compared to control whole embryos. However, the method used to count the cells is not specified. Without that information it is impossible to gauge the accuracy of this measurement.

Thank you for pointing this out. We added the detail method for cell count to materials & methods section. After fixing the embryos, nuclei were stained with DAPI, then we took the confocal images of whole embryos. For avoiding the miscounting, we marked up each nucleus and counted the total number of nuclei on the computer.

Not just the number of cell divisions, but also their orientation may be altered.

Because localized labeling within individual cells is not feasible, we are unable to determine the orientation of each cell. However, since the cells remain attached to each other via cell–cell adhesion, we do not believe that their orientation changes during this process.

c. The morphogenetic transition from flat, to cup and then to sphere is depicted as due to elongation of mostly static cells. However, the supplemental movie provided shows quite a bit of movement of the whole tissue and one can discern some migratory cells (PGCs perhaps?) and some cells that are left out of the sphere. This suggests that there may be cellular rearrangements occurring during this process, which are not normally observed during development of intact embryos. If this were the case, it would have implications also for the hypothesis put forward regarding the mechanisms of antero-posterior axis formation.

The points above could be clarified if the authors analyzed their confocal time-lapses of half embryos to extract cell division events, cell rearrangements and cell death. Alternatively other ad hoc experiments could be devised to address each point.

Thank you for this comment. As suggested, we agree that the cells migrating at the edge of the halved embryo are most likely mesodermal lineage cells. However, based on our detailed observations, we did not detect any dynamic cell movements that traverse the anterior–posterior axis within the embryo. As an example, we include a figure above. In this figure, the top row shows cells at the upper edge of the embryo, the middle row shows cells at the lower edge, and the bottom row shows cells in the central region. Each of these cells was labeled with an asterisk on the computer and tracked over time as development progressed. As the reviewer noted, some cells undergo division. However, since no major cell movements were observed, we believe this strongly suggests that re-formation of the anterior–posterior axis is not driven by large-scale cell rearrangement within the embryo.

2. The morphogenetic events underlying the transition between flat, cup and sphere are unclear.

a. The authors show that the transition to cup is accompanied by excessive elongation of cells, which are transiently longer than cells in whole blastulae. However, the comparison is made only to one blastulation end-point for whole embryos. What do whole embryos cells look like at earlier stages? Can the authors compare with time-matched whole embryos?

Thank you for this comment. We agree that our previous comparison was based on whole embryos at the end-point of development, and thus did not fully capture the apical-basal elongation dynamics of halved embryos. To address this, we conducted new experiments, in which we measured apical-basal length of cell in whole embryos at earlier stages—8 hours (corresponding to the flat stage of halved embryos), 10 hours (cup stage equivalent), and 14 hours (blastula stage equivalent)—and compared these to halved embryos. Our results show that even when compared to whole embryos at the early 8 h stage, the apicobasal length of halved embryos at the cup stage is significantly longer. This finding supports the conclusion that the observed apical-basal elongation is a specific feature of halved embryos. These new data have been added as Supplementary Fig. 3.

b. The authors show an accumulation of actin at the basal side of halved embryos. However the images also show actin signal at the apical side. How much stronger is the signal on the basal side? Is the anisotropy in actin localization similar or stronger than in whole embryos? In other words, is this accumulation a “normal” thing that sea urchin cells do, and that brings about the cupping in halved embryos, or is this a response to the lack of cell-cell contacts on the basal side? The authors suggest it is the former, as they state in the discussion that basal actin accumulation is necessary for blastula formation, but do not provide evidence or a citation for this statement.

Thank you for this insightful and appropriate comment. To address the actin distribution in normal embryos, we have added new data using Lifect-mCherry–expressing embryos. We observed embryos at 8 hours (corresponding to the flat stage in halved embryos) and at 13 hours (during the transition from cup to blastula stages). At 8 hours, we observed almost no actin polymerization on the basal side. At 13 hours, although some accumulation was detectable basally, it was clearly much weaker than the signal observed on the apical side. These findings suggest that the strong basal actin signal observed in halved embryos, which exceeds that of the apical side, is a distinct feature during their blastula formation and may play a critical functional role. We have added Lifect-mCherry images of 13-hour normal embryos to Fig. 2g to illustrate this comparison. Additionally, to maintain consistency in fluorescence representation, the Lifect-mCherry images in Fig. 2h–j have been recolored in magenta.

Regarding the second point about functional significance, we performed inhibitor treatments targeting actin or myosin, based on the hypothesis that unusual basal actin accumulation is important. These treatments suppressed proper blastula formation, supporting the idea that this accumulation plays a crucial role. Ideally, we would manipulate actin polymerization selectively on the basal side while preserving apical actin structure. However, such spatially restricted manipulation remains technically challenging in sea urchin embryos at this time.

c. Is it always the basal side that bends? Do the authors ever observe “inside-out blastulae”?

No, we have never observed that (if it happens, very interesting!). In sea urchin embryos, nuclear position biased to apical side. Based on the nuclear position in live imaging, we concluded that inside-out blastula did not happen.

d. Depolymerization of actin with cytochalasin D affects all actin in the cell, so the authors cannot state that the basal actin accumulation is necessary for the process. They would have to disrupt only the basal actin to make that statement.

As described above, we believe that the irregular actin accumulation observed on the basal side is likely to play a role in the flat-to-blastula transition, as its inhibition disrupts this process. On the other hand, as the reviewer rightly pointed out, we do not have sufficient data to exclude a potential contribution from the apical side, as such targeted manipulation is currently technically challenging. To reflect this, we have revised the text to read: "We also suggest that actomyosin activity at the basal side of cells contributes to the dynamic morphological changes in halved embryos by

facilitating cell shape alterations, although we cannot completely rule out a potential contribution from apical actin."

We have also revised the corresponding text by replacing "indicate" with "suggest" to soften the strength of our claim.

e. Cell division is probably also blocked by treatment with cytochalasin D: can the authors exclude that cell division is necessary for cupping?

We did not examine cell division in detail, but embryos treated with Cytochalasin D from the flat stage were still able to reach the cup stage. This suggests that cell division is not essential for the flat-to-blastula transition, at least up to the cup stage, as highlighted by the reviewer. However, we cannot completely exclude the possibility that cell division becomes necessary during later stages of this transition. Nevertheless, based on our additional experiments with Blebbistatin and Tetraspanin inhibition, we believe that cell adhesion and cytoskeletal modifications play a more critical role than cell division in driving the flat-to-blastula transition.

d. The tetraspanin morpholino experiment is quite striking, although several controls are missing. Is cell division impaired? Is apoptosis induced? Do these embryos produce a normal hyaline layer?

Thank you for this comment. To address both questions regarding cell division and apoptosis, we counted the number of cells in tetraspanin morphants. For this experiment, we cultured the morphants approximately two hours longer than the controls—long enough for control embryos to reach the blastula stage—to confirm that the morphants still remained in the flat stage.

We also confirmed that, despite the extended culture period, the morphants had a cell number comparable to that of control embryos transitioning from the sphere to the blastula stage, yet they still maintained a flat morphology (see figure above). During this observation, we frequently detected cells undergoing mitosis by DAPI staining, and the total cell number was not significantly reduced compared to controls. These results

suggest that apoptosis is unlikely to be a major factor contributing to the sustained flat morphology in the morphants.

Regarding the hyaline layer, we considered it important to determine whether tetraspanin knockdown affects its formation in the context of this question. To address this, we examined the hyaline layer in tetraspanin morphants. As shown in the figure above, the hyaline layer was clearly present even in the morphants. Therefore, we believe that the persistence of the flat morphology in tetraspanin morphant halved embryos is not due to defects in hyaline layer formation.

3. Stronger evidence is needed to support the proposed mechanism for anteroposterior axis formation. The authors bring forward a model in which flat embryos have an anteroposterior axis, equivalent to the one that would have formed if the blastomere had not been isolated, that gets re-organized upon closure of the cup into a sphere. They suggest that the original anterior-most and posterior-most ends of the half embryo meet once the cup is closed and that the axis is reset at that point, due to a reactivation of Wnt/b-catenin signalling. The

evidence provided in support of this model is enticing, but not quite compelling. In particular,

a. To show that the anterior-most and posterior-most cells of the half-embryo meet when the cup closes, the authors use HCR for anterior and posterior genes. While they beautifully show the domains coming closer and then separating again after closure of the sphere, this technique does not allow distinguishing between changes in gene expression and cell movement. It is therefore unclear whether cells that originally expressed anterior and posterior genes come into contact and then change their expression profile, or if they move, before or after closure of the cup.

Also, the HCR on fixed samples at the sphere stage do not provide information on the position of the FoxQ2 and Axl1 expression domains with respect to the point of cup closure.

To clarify these points the authors could perform a double tracing experiment, marking one anterior and one posterior cell with dyes of different color and then follow their relative position at cup, early sphere, late sphere and gastrula. Images for the same half embryos could be taken at multiple time points. Alternatively, they could take time lapses of halved embryos and track anterior and posterior cells across flat, cup and sphere stages.

Thank you for this comment. We had the same question in mind during the course of our study, which led us to perform the experiments shown in Fig. 3g,h and Supplementary Fig. 7. If the anterior–posterior (A–P) axis were re-established through cell rearrangement—specifically, by the original anterior or posterior pole cells migrating to re-form the axis—then our cell lineage tracing results should show that anterior pole cells are always located at the anterior side of the embryo, and posterior pole cells at the posterior side, with 100% consistency relative to the position of the archenteron. However, our data do not support this outcome. The results we present argue strongly against A–P re-establishment being driven by directed migration or rearrangement of these original pole cells, suggesting instead that other mechanisms are at play.

b. The authors show that nuclear b-catenin is present in halved embryos, and state that it is not normally active in whole embryos at equivalent stages. However, domains of

nuclear b-catenin have been described at blastula stages in other urchins (see Logan, 1999) and whether the sphere stage corresponds to blastula stages in which some cells may still have nuclear b-catenin in whole embryos is unclear. The authors should provide similar data on b-cat localization for time matched whole embryos.

c. The authors state that nuclear b-catenin signal is observed in a domain adjacent to the original posterior cells: how was this determined?

We would like to address points b and c together. Thank you for the comment. As noted in the paper by Logan et al., which the reviewer kindly pointed out, nuclear localization of β -catenin is observed in a portion of the vegetal half in normal embryos at this stage. Consistent with this, in our current halved embryo experiments (Supplementary Movie and Fig. 3j–l), the lower half of the sealed embryo shows green signal—although it may be difficult to see clearly, and we apologize for that. This region corresponds to the original posterior, indicating that β -catenin in this region has not been degraded to the same extent as in the original animal half. To avoid confusing, we deleted “, not observed in intact embryos,” from the text.

Following sealing, we observed a transient increase in nuclear β -catenin activity in cells adjacent to the original posterior region, suggesting that at least some neighboring cells acquired β -catenin activity. In contrast, in whole embryos at the corresponding stage,

nuclear β -catenin is restricted to a very small area at the vegetal pole (attached figure above).

d. To experimentally prove the need for Wnt/b-catenin signalling to re-organize the axis, the authors analyze halved embryos injected with $\Delta Ecad$. The experiment is not well described, but we assume that the zygotes were injected and then blastomeres were separated at 2-cell stage. In this case, all Wnt/b-catenin signalling would be impaired, meaning these embryos would have failed to generate any axis, and not only to re-organize it upon closure. A more convincing experiment would be blocking Wnt/b-catenin signalling after the primary axis has been established, which could be achieved using inhibitors like C59. This should result in embryos that have anteriorly fated cells (expressing FoxQ2 or making an apical tuft) right next to posterior invaginating cells.

Thank you very much for this excellent suggestion. We were also interested in this question and had previously attempted to inhibit canonical Wnt (cWnt) signaling using inhibitors such as iCRT and XAV. However, we were unable to achieve complete inhibition of cWnt activity even in normal embryos, so we had initially set this approach aside. We were not aware of C59 until the reviewer's suggestion, and we subsequently tested it.

First, we examined its effect in normal embryos. Although C59 did not block mesodermal cell ingression in *H. pulcherrimus*, we observed an expansion of the apical tuft region (i.e., the anterior neuroectoderm), suggesting that C59 can at least partially inhibit cWnt signaling in this species.

We then applied C59 to halved embryos starting from the cup stage and examined whether the anterior marker *foxQ2* and the posterior marker *foxA* were located at opposite poles after sufficient time had passed in the blastula stage. In control (DMSO-treated) halved embryos, *foxQ2* and *foxA* were positioned at opposite ends in 100% of cases, indicating successful A–P re-formation. In contrast, nearly half of the C59-treated halved embryos failed to show polar localization of *foxQ2* and *foxA*.

Although C59 may not completely block cWnt activity, these results strongly suggest that cWnt signaling becomes transiently active during the cup-to-blastula transition and is necessary for A–P axis re-establishment. We have now included these findings in the main figures (see Fig. 3m–n'), and have moved the data involving non-canonical Wnt inhibitors to the Supplementary Fig. 8. Thank you again for this valuable suggestion.

e. If the contact between anterior and posterior domains in the sphere stage is necessary to trigger axis reorganization, the axis should stay the same if the contact were to be prevented. Can the author keep the halved embryos flat and check whether the axis remains as initially set or re-organized?

This is an excellent perspective. We also considered the possibility of inhibiting cell–cell contact only, without interfering with overall development. However, we were unable to identify a feasible experimental approach to achieve this at present. Therefore, the question of what exactly is triggered by cell–cell contact to induce the reactivation of cWnt signaling remains open, and we hope to address this in a future study.

Minor concern:

1. In the discussion the authors argue that the mode of cup and sphere formation is passive, as opposed to an active boundary recognition mode. What is meant with these two terms, passive and active, is very unclear. Cells elongate and curve the tissue to close it up, which does not seem a passive mechanism.

Thank you for this comment. In this study, we initially considered the possibility that halved embryos might rely on active interactions between cells within the embryo to organize themselves into a spherical blastula. However, our findings suggest otherwise: the flat-to-blastula transition appears to be achieved through the accumulation of simple, cell-intrinsic behaviors—such as adhesion and elongation—within individual cells, rather than through coordinated, intercellular signaling. To reflect this interpretation

more clearly, we have substantially revised the final part of the first paragraph of the Discussion, avoiding the use of the term "passive" and instead emphasizing the self-regulatory nature of individual cellular behaviors.

2. In the discussion, several parallels are drawn between dissociated sea urchin and sea star cells, which ignore the fact that sea star embryos do not have a hyaline layer and become quite flattened even just after removal of their fertilization envelope: it is not surprising that sea star cells would not form a compact cluster upon blastomere separation. The reasons why sea star blastomeres form a flat sheet upon dissociation may be entirely different from the reasons why sea urchin cells do the same.

We agree with this reviewer. We deleted the corresponding sentence.

“Similarly, isolated sea star blastomeres also go through a flat stage before forming a blastula^{5,13}, suggesting that this may be a conserved feature across echinoderms.”

Reviewer #4 (Remarks to the Author)

Thank you so much for your help in improving our manuscript.

REVIEWERS' COMMENTS

Reviewer #1 (Remarks to the Author):

The authors have answered most of the questions and concerns I had. the only point that I think remains unclear is when the authors discuss about the role of FoxQ2. They argue that the completion of FoxQ2 expression to the anterior end is essential for initiating D/V axis formation. However, FoxQ2 function is not required for D/V axis formation in the unperturbed embryo and nodal expression is normal in the FoxQ2 morphants. Wouldn't we expect nodal to be expressed prematurely in the absence of this repressor of nodal expression? There is something that I don't get here. Otherwise, they have clarified all the other points that were too vague and they have provided numerous supplementary data and additional analyses that further increase the interest of the paper.

Thank you for this helpful comment. We agree that our previous manuscript did not clearly describe the relationship between FoxQ2 and nodal. Previous studies have shown that FoxQ2 does not significantly affect the initiation of nodal expression, but rather suppresses its autoregulatory loop. We have revised the text on pages 16–17 to more accurately reflect this point.

We also appreciate your positive feedback on the other revisions we made.

Reviewer #2 (Remarks to the Author):

The authors have addressed all the points I raised. I congratulate the authors on this thorough and elegant work!

Your encouraging comments are greatly appreciated.